# Assessing Wood and Soil Carbon Losses from a Forest-Peat Fire in the Boreo-Nemoral Zone

Andrey Sirin [1,*], Alexander Maslov [1], Dmitry Makarov [1], Yakov Gulbe [1] and Hans Joosten [2]

1    Institute of Forest Science, Russian Academy of Sciences, Moscow Region, 143030 Uspenskoye, Russia; amaslov@ilan.ras.ru (A.M.); scorpionstigr@mail.ru (D.M.); goulbe@ilan.ras.ru (Y.G.)
2    Institute of Botany and Landscape Ecology, Greifswald University, Partner in the Greifswald Mire Centre, Soldmannstrasse 15, D-17487 Greifswald, Germany; joosten@uni-greifswald.de
*    Correspondence: sirin@ilan.ras.ru; Tel./Fax: +7-495-634-5257

**Abstract:** Forest-peat fires are notable for their difficulty in estimating carbon losses. Combined carbon losses from tree biomass and peat soil were estimated at an 8 ha forest-peat fire in the Moscow region after catastrophic fires in 2010. The loss of tree biomass carbon was assessed by reconstructing forest stand structure using the classification of pre-fire high-resolution satellite imagery and after-fire ground survey of the same forest classes in adjacent areas. Soil carbon loss was assessed by using the root collars of stumps to reconstruct the pre-fire soil surface and interpolating the peat characteristics of adjacent non-burned areas. The mean (median) depth of peat losses across the burned area was $15 \pm 8$ (14) cm, varying from $13 \pm 5$ (11) to $20 \pm 9$ (19). Loss of soil carbon was $9.22 \pm 3.75$–$11.0 \pm 4.96$ (mean) and 8.0–11.0 kg m$^{-2}$ (median); values exceeding 100 tC ha$^{-1}$ have also been found in other studies. The estimated soil carbon loss for the entire burned area, 98 (mean) and 92 (median) tC ha$^{-1}$, significantly exceeds the carbon loss from live (tree) biomass, which averaged 58.8 tC ha$^{-1}$. The loss of carbon in the forest-peat fire thus equals the release of nearly 400 (soil) and, including the biomass, almost 650 tCO$_2$ ha$^{-1}$ into the atmosphere, which illustrates the underestimated impact of boreal forest-peat fires on atmospheric gas concentrations and climate.

**Keywords:** forest fire; peat fire; climate change; GHGs; carbon dioxide



## 1. Introduction

Catastrophic fires in forest ecosystems cause huge emissions of greenhouse gases (GHGs), especially carbon dioxide, into the atmosphere [1]. They are facilitated by climate change and associated warmer and drier conditions [2] with positive feedback on climate change. About 10% of global fire carbon emissions are attributable to fires in boreal and temperate forests [3,4]. In areas with a concentration of population and economic structures, their environmental impact is proportionally even larger; this is especially true for forest-peat fires, i.e., forest fires in which also the peat soil burns. Such fires may penetrate deep into the peat soil and kill the forest stand by damaging the tree roots [5]. Lower groundwater levels create deeper fires, more damage to trees, and larger soil carbon losses [6], exacerbating the effects of climate change [7]. Peat fires contribute substantially (in some years up to 15% [8]) to global anthropogenic GHG emissions [9–13].

Smouldering combustion may cause peat fires to burn for long periods, even during prolonged periods of rainfall and snow cover [7,9,14,15]. Smouldering combustion produces not only CO$_2$ but also organic volatiles, which are dangerous to human health [14,16]. These effects were especially evident during the catastrophic forest-peat fires in central European Russia in July and August 2010 [17–19], when a combination of anomalous hot weather [20] and extreme smog [21] dramatically increased excess mortality [22,23].

Forest-peat fires are mostly associated with Southeast Asia [24], which may have had planetary consequences in some years [25,26]. However, peat fires may occur everywhere peatlands exist [14], but most often in the boreal zone [27]. More than 21% of the territory

of Russia is covered by peat, i.e., 139 million ha of peatland with ≥30 cm of peat and 230 million ha of paludified land with <30 cm of peat [28]. In its European part, the proportion is 17% [29]. Moreover, 38% of peatlands and 47% of paludified lands in Russia are covered with forest or sparsely treed vegetation [30].

While recognizing their impact on climate, the Intergovernmental Panel on Climate Change (IPCC 2014) stresses the methodological problems of accounting for peat fire emissions and the lack of data on carbon losses from peat fires [12]. Compared to the tropics, few data are available on emissions from peat fires outside the tropics, especially in the boreal zone, although the frequency of the latter may be higher, their duration longer, and their consequences more serious [7].

In order to estimate carbon losses from forest-peat fires, the soil carbon losses from peat combustion must be added to the losses from biomass burning. The existing methods to assess soil carbon losses are based on differences in ash concentrations between burned and unburned soil horizons [6,31], on comparing pre- and post-fire surface heights using multitemporal LiDAR data [32], and on reconstructing the pre-fire surface from the position of the root collar of trees [10,33]. The latter approach is the most appropriate for forest-peat fires.

This study analyses biomass and peat carbon losses and consequent carbon dioxide emissions from a forest-peat fire as part of a project to assess the consequences of the catastrophic fires around Moscow in 2010 and the effectiveness of peatland rewetting to prevent further peat fires [19]. Pre-fire forest cover was reconstructed using multispectral high-resolution satellite imagery and ground surveys of the surrounding forests. From the position of the root collar of preserved tree stumps in the burned area, we reconstructed the original pre-fire peat surface by comparing the peat stratigraphy of the burned and adjacent unburned areas. This strategy allowed us to assess the fire-associated loss of soil carbon.

## 2. Materials and Methods

### 2.1. Study Area

The study area is located in the paludified eastern part of the Moscow region (Figure 1), in Shatura district in the western part of the Meshchera lowland (55°37′38.75″ N, 39°34′32.50″ E). Moscow region, also recognized as the 'subject' of the Russian Federation surrounding Moscow city, has an area of 44,329 km$^2$ and is located in the boreo-nemoral (mixed coniferous-broad-leaved) forest zone [34]. Peatlands cover over 250,000 ha or 6% of Moscow region [35] (Figure 1). Shatura is the coldest district in the Moscow region. Overcast weather is typical; clear days occur only 1–2 times a month. Average annual air temperature is +3.6 °C, with an average January temperature of −11 °C, and an average July temperature of +17.6 °C. Average daily temperature is above 0 °C for 210–220 days of the year. Average annual precipitation is 524 mm (450–800 mm), with two out of three rainfalls in the warm period of the year. The duration of snow cover is 150–155 days.

The territory is relatively flat and heavily paludified, with lakes occupying the major depressions. Many peatlands in this region have been drained for peat extraction, agriculture and forestry [36,37]. Furthermore, the study area has in part been drained, probably in the first half of the 20th century. In the eastern part of the burned area, along the road and in the unburned forest to the south, drainage ditches are still visible. To the south and north, the burned area is bordered by a heterogeneous forest with a predominance of birch (*Betula pendula*, *Betula pubescens*), aspen (*Populus tremula*), alder (*Alnus glutinosa*), Scots pine (*Pinus sylvestris*), and some rowan (*Sorbus aucuparia*) in the understorey. To the east, the site is bordered by an automobile road and to the west by a lake.

In the exceedingly dry summer of 2010, a 9-ha large forest area was fully destroyed by a fire that started as a ground fire but rapidly developed into an underground (peat) and partially crown fire. The fire was first recorded in a QuickBird image from 25 July 2010 (Figure 2a), and by mid-August, the fire was already extinguished (Landsat 5 image from 18 August 2010). Ikonos imagery from 12 June 2011 shows abundant fallen, partially

burned tree trunks at the burned site (Figure 2b). At the start of our study (2013), the burned area was already overgrowing; the dead stems and trunks had been removed with only some dead stems and stumps remaining [38].

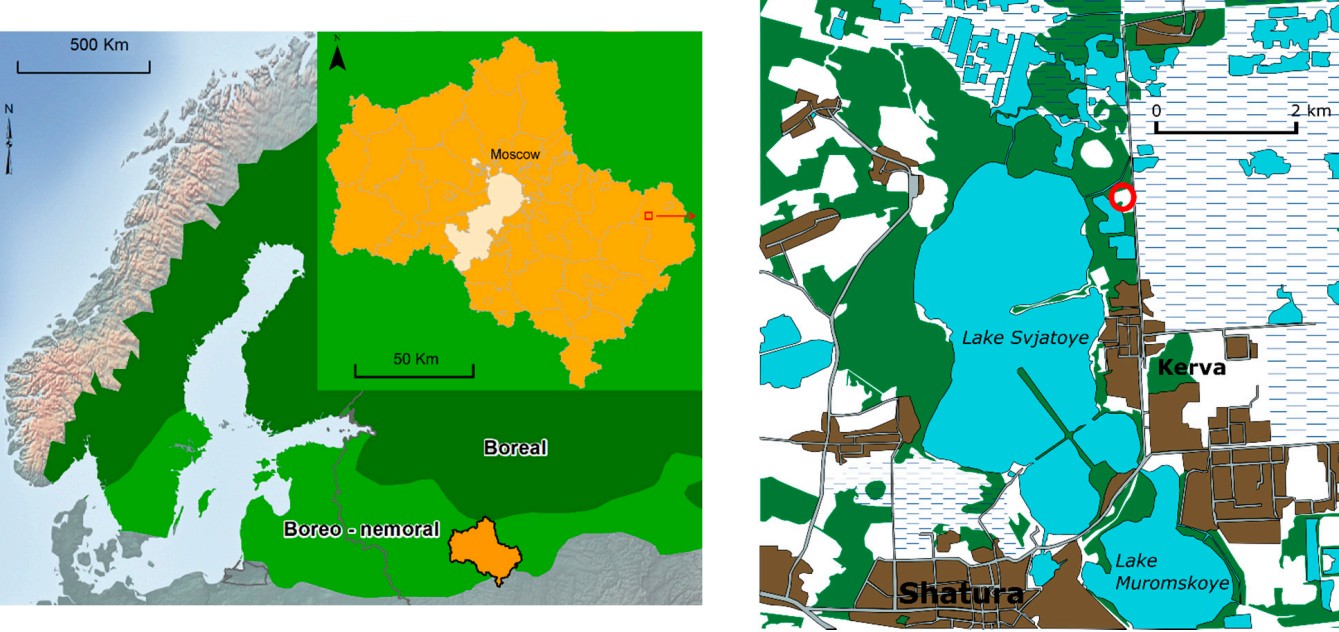

**Figure 1.** Location of the study area. The red circle in the right figure shows the location of the burned site studied. In the right figure: green, forest; blue, water; brown, urban area (incl. dacha complexes); white, arable land on mineral soil; striped, open peatland.

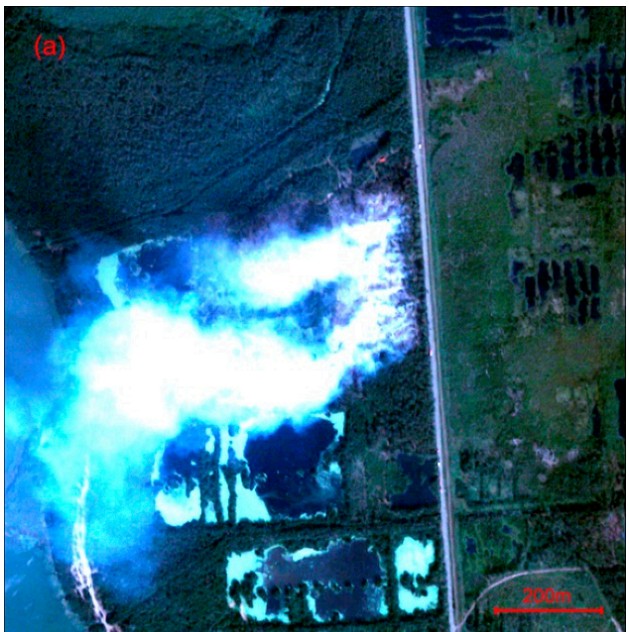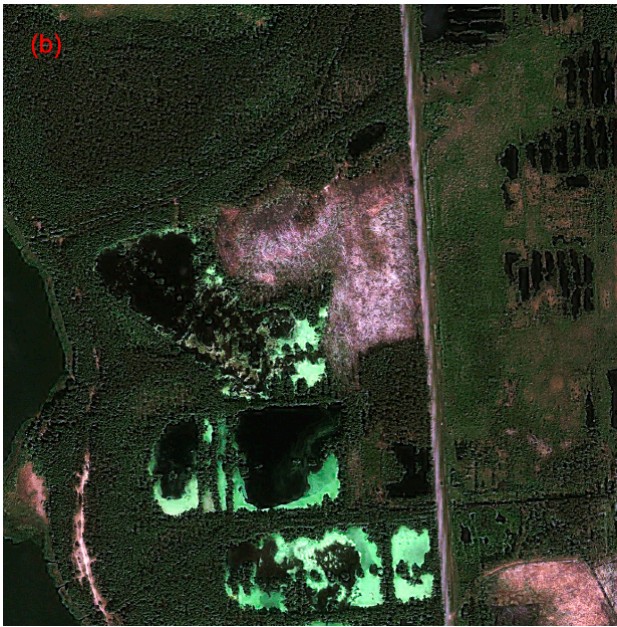

**Figure 2.** Active fire location, QuickBird image, 25 July 2010, © 2021 DigitalGlobe, Inc. (**a**); burned area with fallen trees on Ikonos image, 12 June 2011, © 2021 GeoEye, Inc., (**b**).

### 2.2. Reconstruction of Forest Vegetation

In Russia, forest inventory data are usually used to estimate stemwood losses after a fire. However, as the burned area is located outside the State Forest Fund lands, no forest inventory data were available, and characteristics of the pre-fire stands had to be

reconstructed. We used pre-fire remote sensing imagery, classification techniques, and ground truth data according to the following steps: Selection of a pre-fire satellite image with the required resolution; classification of the image, including the burned area and a sufficiently large adjacent zone, assuming that the burned and adjacent areas had the same forest types; creation of a thematic vegetation map based on the obtained thematic classes (homogeneous in terms of spectral pixel characteristics); selection and description of ground truthing plots outside the burned area in the thematic classes present in the burned area before the fire; calculation of wood and carbon stock per hectare for each class based on the information from relevant ground plots; calculation of wood and carbon stocks for the entire burned area based on the area share of each class.

### 2.3. Classification of the Pre-Fire Satellite Image

We used a high-resolution (10 m) pre-fire Spot-5 multispectral image and ScanEx Image Processor software [39] to produce an ISODATA unsupervised land type classification of the 2010 burned area polygon and an adjacent unburned zone of 250–300 m wide. A raw image with the original channel histograms and original pixels provided the following classification parameters as input data: 4 spectral channels, 20 thematic classes, and 30 iterations. The classification results were saved to a raster layer, after which automatic vectorization of the thematic polygons was conducted for further analysis using GIS tools in MapInfo.

### 2.4. Ground Truthing and Assessment of Biomass Loss

The land type thematic classes were preliminarily identified using vegetation relevés within the map units of each thematic class in the zone adjacent to the burned area. At each point, we documented our progress with photos and brief descriptions of the land type and vegetation. Several forest inventory plots (100–400 m$^2$ depending on stand density) were established for each forest thematic class present in the burned area before the fire. Diameter at Breast Height (DBH) and the height of all trees and saplings with DBH > 3.2 cm (10 cm perimeter) were recorded, and the age of a representative selection of trees was determined using an increment borer. The growing stock volumes for birch, pine, and aspen in the forest inventory plots were calculated using standard volume tables [40], and for alder and rowan using regression equations. A conversion factor of 0.50 was used to convert biomass (dry weight) to carbon equivalents (C) (ton).

The diameter of well-preserved stumps in the burned area was measured, and the tree age determined using saw cuts in order to verify the thematic classes obtained in the Spot-5 image classification. Some stumps had been crushed by heavy machinery when removing the remaining wood from the burned site, while others must have burned completely. Aspen stumps seemed to have suffered most because they have lower mechanical strength than other species.

### 2.5. Determining Peat Burning Depth

The pre-fire surface of the burned area was reconstructed using root collar heights of the remaining tree stumps [10]. This method is less dependent on the elapsed time since the fire, in contrast with the assessment of soil carbon losses based on differences in ash concentrations [6,31]. The stumps were spatially referenced using a GPS receiver. The burned-out area surface was levelled at 2 m intervals along seven transects crisscrossing the entire area with approximately right angles rather evenly. These transects were also used to study peat stratigraphy using a Russian Peat Sampler (chamber corer). Additionally, the thickness of the peat deposit was determined using a peat dipstick.

Local peat loss was determined by measuring the distance of the root collar for each stump in the burned area to the new peat surface at 4–5 points and c. 50 cm distance around the stump (Figure 3 left). The mean burn depth in the local microrelief was determined using 2–3 stumps. Based on the interpolation of data from 306 tree stumps, a regular grid of 1 × 1 m peat burning depths was created as well as a map of lines with equal peat

burning depths. The reconstructed pre-fire surface and the burn depth distribution were used for calculating the volume of peat loss. As no stumps could be retrieved from there, the northwestern part of the burned area was excluded from further analysis, reducing the studied burned area to 8.1 ha.

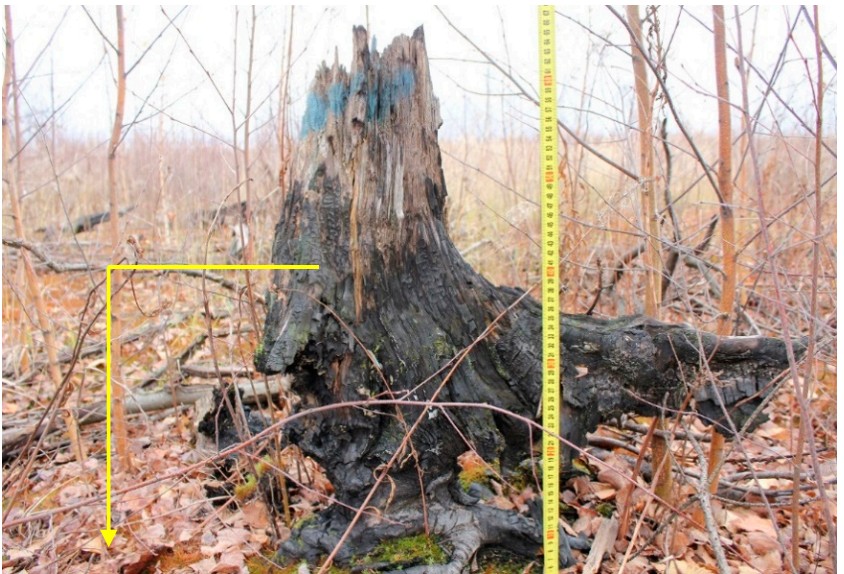 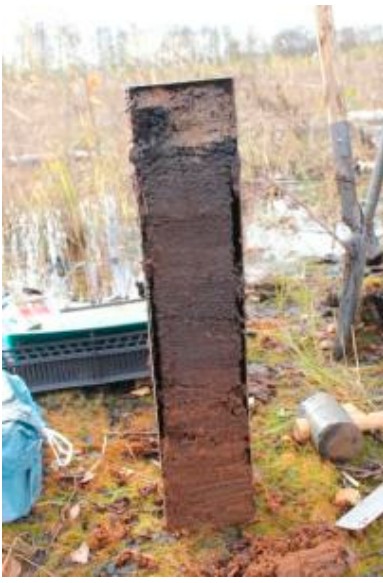

**Figure 3.** Reconstruction of the pre-fire surface level using the root collar of trees (**left**) and peat cores to assess peat bulk density and other soil variables (**right**).

In the burned area, stumps of birch, aspen, pine, and black alder were found. As for different tree species and growing conditions, the position of the root collar relative to the peat surface may differ; we determined this position for different tree species in forests bordering the burned area.

Data processing was performed in Excel, Surfer 11 Golden Software (surface construction), and MapInfo GIS (spatial information).

### 2.6. Peat Characterisation and Determination of Soil Carbon Losses

Peat characteristics up to a depth of 50 cm were determined layer-by-layer on eight 5 × 5 m peat inventory plots (Figure 4) in the burned area (plots 3, 4, 7 and 8) and on the adjacent unaffected forest area (plots 1, 2, 5 and 6). Plot 1 is dominated by birch up to 20 m and pine up to 22 m high; plot 2 by birch up to 27 m and pine up to 27 m high; plot 5 only by alder up to 20 m high; plot 6 (directly bordering the burned area) by birch and aspen up to 25 m high.

Peat samples were taken using a U-shaped, 20 cm wide and 11 cm deep stainless steel box, which cuts with its sharpened edges into the sidewall of a pit (Figure 1 right). The extracted monolith was cut into layers of 5 cm thick, of which the botanical composition and the degree of decomposition were determined microscopically using the centrifugation method. Bulk density ($D$) was determined in two parallel samples from the same slices after drying at 105 °C; ash content ($A$) by calcination at 800 °C; and $C$ content by analysing two 2–3 g subsamples of dry crushed peat with a varioMICRO cube elemental analyser (Elementar, Germany) in threefold repetition.

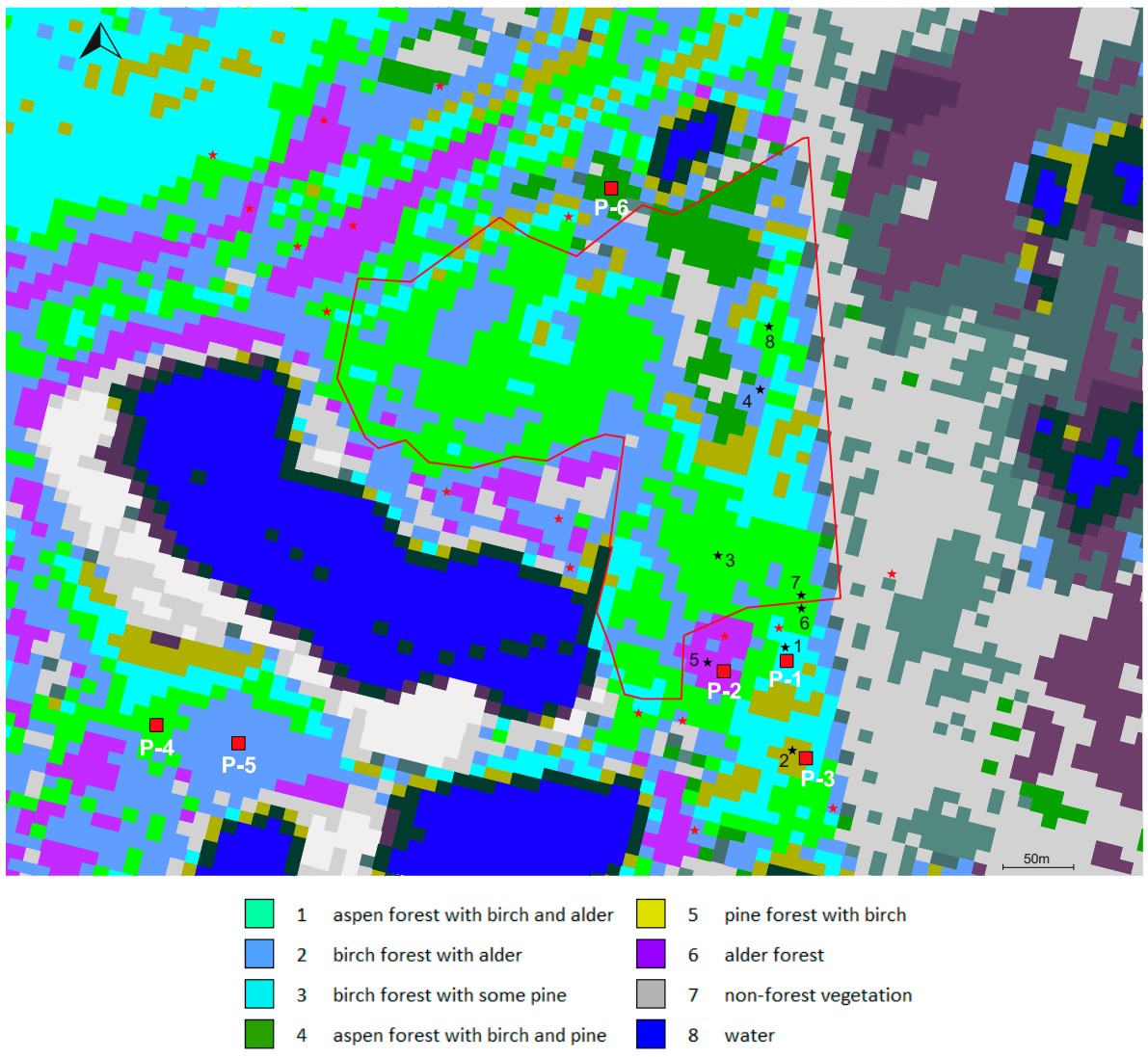

**Figure 4.** Results of automatic classification of the pre-fire Spot 5 image from 3 June 2007 and location of vegetation relevés (red asterisks), forest inventory plots (P-1 to P-6), and peat inventory plots (black asterisks, 1–8). The solid red line shows the boundaries of the area burned in 2010.

## 3. Results

### 3.1. Biomass and Biomass Carbon Losses

Table 1 lists the forest thematic classes and the aboveground carbon losses due to the death of tree stands and understorey. Detailed calculations and stand characteristics are given in the Supplementary Materials (Tables S1 and S2). Aboveground biomass carbon losses averaged 62.9 t ha$^{-1}$ for all forest thematic classes and 58.8 t ha$^{-1}$ when including non-forest classes.

### 3.2. Reconstruction of Pre-Fire Vegetation

Our first attempts to use Landsat-7 data for classification were unsuccessful due to the small size of the object (380 × 440 m) and the pixel size of the Landsat image (30 × 30 m) being too large to detect the smaller forest map units. We finally obtained the multispectral cloudless Spot-5 image from 3 June 2007 with a pixel size of 10 × 10 m for the thematic classification. Figure 4 shows that the small pixel size of Spot 5 can identify several thematic classes (homogeneous areas of vegetation) within the future burned area. More importantly, the same classes are also present in the adjacent area, which allowed us to characterize these classes using relevés and plots in that area after the fire.

As shown by the ground surveys, the largest portions of the burned area had been covered by birch and aspen forests of various ages, in some places intermixed with alder or pine. Small fragments were occupied by grass and shrub communities, the waterlogged depressions with sparse birch and alder or pure stands of young alder. In the adjacent zone (outside the burned area) bogs, fens, peatlands overgrown after the fires of 2002, ditches, lakes, and areas with bare peat were also described. Six main pre-fire forest thematic classes were identified by combining the most similar thematic classes. All non-forest thematic classes were combined into one (Figure 4).

**Table 1.** Characteristics of forest thematic classes and aboveground carbon losses as a result of tree stand and understorey death after fire. Non-forest classes were not included in the calculations.

| Thematic Class | Plot | Thematic Class Description | Area (ha) | C (t ha$^{-1}$) | C (t) |
|---|---|---|---|---|---|
| 1 | 4 | Aspen forest with birch and alder | 3.56 | 91.18 | 324.60 |
| 2 | 5 | Birch forest with alder | 2.45 | 20.18 | 49.44 |
| 3 | 1 | Birch forest with some pine | 1.19 | 96.62 | 114.98 |
| 4 | 6 | Aspen forest with birch and pine | 0.68 | 27.91 | 18.98 |
| 5 | 3 | Pine forest with birch | 0.53 | 39.32 | 20.84 |
| 6 | 2 | Alder forest | 0.03 | 72.59 | 2.18 |
| Total: | | | 8.44 | 62.9 | 531.02 |

### 3.3. Burned-Out Depth of Peat

Figure 5 shows the map of peat burn depth isolines. The levelling data and the maps of pre-and post-fire surface heights of the burned area are presented in the Supplementary Material (Figure S3). Mainly for alder, the results show that the position of the root collar in pre-fire conditions should be corrected, but because of its small representation in the burned area, such correction was not made.

Peat thickness after the fire is 0.9 m on average, with a maximum thickness (1.5 m) being reached in the centre of the site and the forest beyond the southern boundary; the thickness decreases towards the northern part of the burned area with mineral soil surfacing in places. Height difference within the burned area reaches almost 2 m, with surface elevation rising from southwest to northeast. Isolated local elevations occur in the centre of the burned area, and at the start, and along the contours of the protruding westward section. Burn depths generally increase with the increasing height of the reconstructed (pre-fire) soil surface, which can be explained by elevated areas being better drained and more exposed to fire.

The burn depth varied considerably (Figure 2, see also Supplementary Material Figure S4). Average values from 13 to 15 cm were characteristic for all areas, except for 'aspen forest with birch and pine' where the burn depth was 20 ± 9 cm (Table 2). With 11–15 and 19 cm, respectively, the median burning depth values were slightly less than the mean values, a similarity that may indicate the symmetric distribution of peat burn depth values. At the same time, burn depths differed for different forest stands (Supplementary Material Figure S4), with mean and median values for 'aspen forest with birch and alder' being 13 ± 8 and 12 cm; for 'pine forest with birch' 14 ± 5 and 13 cm; for 'birch forest with some pine' 15 ± 6 and 14 cm; for 'birch forest with alder' 15 ± 8 and 15 cm; and for 'aspen forest with birch and pine' 20 ± 9 and 19 cm, respectively. Maximum peat burn depth values reached 50 cm or more, particularly in the 'aspen forest with birch and pine', which also had the highest average values.

### 3.4. Soil Carbon Loss

Peat botanical composition, degree of decomposition (*R*), ash content (*A*), bulk density (*D*) and carbon content (*C*) were obtained from eight soil pits (Supplementary Material Table S3) and the carbon stock calculated for each 5cm thick peat layer (Figure 6, Supplementary Material Table S3). The sample pits comprised burned and unburned areas of similar pre-fire tree cover (Figure 2), with the pits at plot 6 (outside the burned area)

and plot 7 (inside) being located adjacent to each other on either side of the fire spread boundary. Considering the small data scatter (especially at shallow depths, which account for most of the peat burning), we used the same carbon stock to depth relation for all pits and used this relationship to calculate the depth-dependent carbon loss associated with peat soil burning (Table 2).

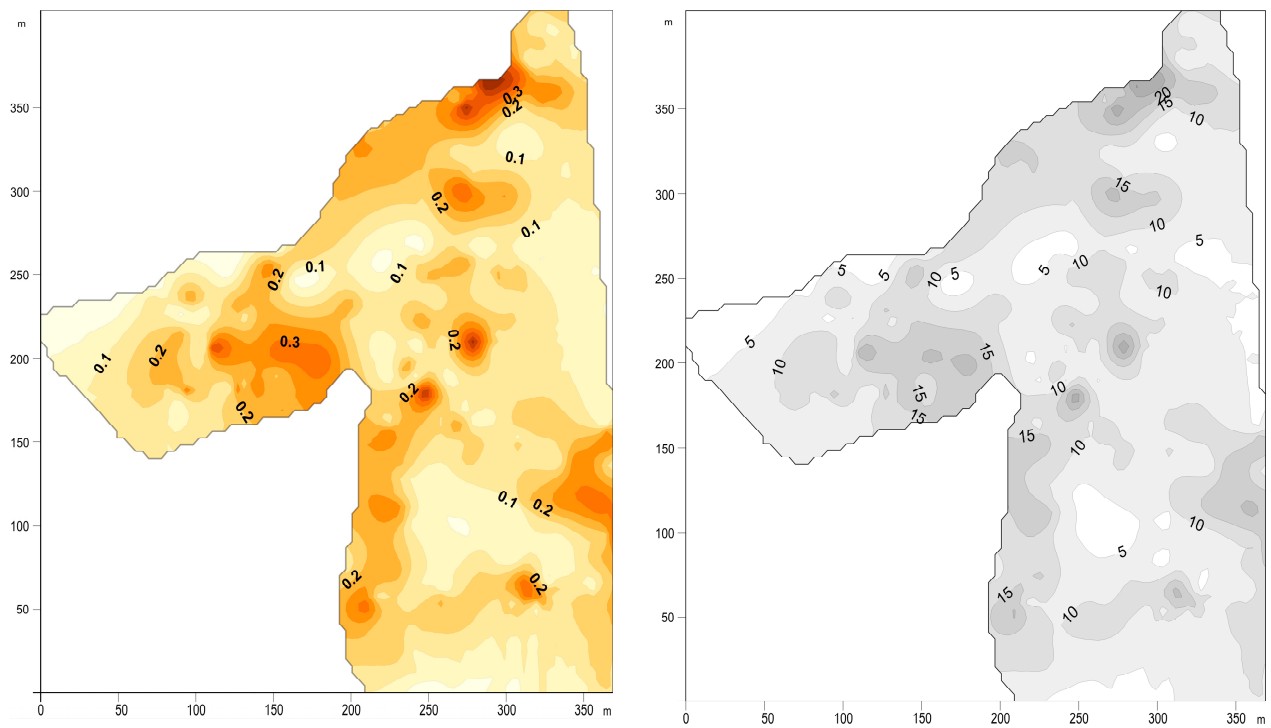

**Figure 5.** Peat burn depths (m) (**left**) and loss of soil carbon (kg m$^{-2}$) (**right**).

**Table 2.** Peat burn depths and loss of soil carbon in different thematic classes. *M*, arithmetic mean; *S*, standard deviation; *Me*, median.

| | Pine Forest with Birch | Birch Forest with Some Pine | Aspen Forest with Birch and Pine | Aspen Forest with Birch and Alder | Birch Forest with Alder | Alder Forest | Non-Forest | Total Burned Area |
|---|---|---|---|---|---|---|---|---|
| | | | | Area | | | | |
| ha | 0.48 | 0.98 | 0.67 | 3.11 | 2.25 | 0.08 | 0.51 | 8.08 |
| % | 6.0 | 12.1 | 8.3 | 38.5 | 27.8 | 1.0 | 6.3 | 100 |
| | | | | Peat burn depth, cm | | | | |
| *M* | 14 | 15 | 20 | 13 | 15 | 13 | 17 | 15 |
| *S* | 5 | 6 | 9 | 8 | 8 | 5 | 7 | 8 |
| max | 32 | 32 | 50 | 50 | 51 | 27 | 35 | 51 |
| min | 3 | 1 | 6 | 0 | 0 | 5 | 4 | 0 |
| *Me* | 13 | 14 | 19 | 12 | 15 | 11 | 17 | 14 |
| | | | | Loss of soil carbon, kg m$^{-2}$ | | | | |
| *M* | 9.22 | 9.8 | 12.9 | 8.61 | 9.8 | 8.61 | 11.0 | 9.8 |
| *S* | 3.75 | 4.35 | 6.18 | 5.57 | 5.57 | 3.75 | 4.96 | 5.57 |
| max | 20.2 | 20.2 | 31.1 | 31.1 | 31.7 | 17.1 | 22.0 | 31.7 |
| min | 2.53 | 1.31 | 4.35 | 0 | 0 | 3.75 | 3.14 | 0 |
| *Me* | 8.61 | 9.22 | 12.3 | 8.0 | 9.8 | 7.39 | 11.0 | 9.22 |

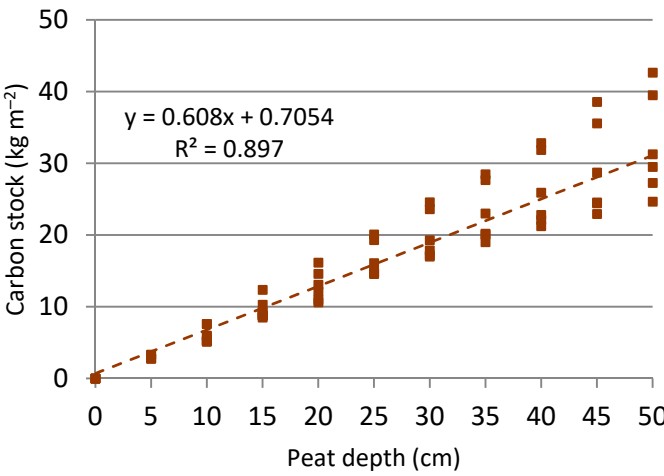

**Figure 6.** Soil carbon stock vs. peat layer thickness.

The highest soil carbon loss (Table 2) was found for 'aspen with birch and pine' (mean $12.9 \pm 6.18$, median 12.3 kg m$^{-2}$). In this class, similar to 'aspen forest with black alder' and 'birch forest with black alder', carbon losses reached local values of 30 kg m$^{-2}$ and higher, whereas in other classes, the losses usually remained under 20 kg m$^{-2}$. The lowest losses (mean $8.61 \pm 3.75$, median 7.39 kg m$^{-2}$) were found for 'alder forest'. In the other classes, losses varied from $9.22 \pm 3.75$ (mean) and 8.0 (median) to $11.0 \pm 4.96$ and 11.0 kg m$^{-2}$. The average soil carbon loss for the entire study area was $9.8 \pm 5.57$ with a median of 9.22 kg m$^{-2}$, which equals 98 and 92 tC ha$^{-1}$, respectively.

## 4. Discussion

Remote sensing images prior to and after the fire are often used to detect wildfire damage [41,42]. In this study, we used pre-fire imagery to estimate biomass and carbon losses by finding identical forest stands in and adjacent to the burned area (unburned ecosystem analogue approach). Alternative methods include using recent forest inventory data (which were not available in our case and most cases) or a detailed survey of burned trunks (which had been cut down and removed by the start of our work). Estimating tree stand carbon losses from preserved tree stumps may lead to severe underestimation; before the 2010 fire, the total number of birch, aspen, pine, and alder trunks in the area was 23,500 (cf. Table 2 and Table S1), of which after the fire and the cutting of the dead stands, only 368 stumps and dead trunks from unburned and larger trees were left (Figure S2).

Even fewer (only 306) stumps could be used for reconstructing the pre-fire soil surface level from the position of their root collar. However, this was sufficient to determine the depth of peat burning in the entire burned area and also reveal spatial patterns. This method is probably the most effective to estimate soil carbon loss from forest-peat fires [10,33]. The 'excess ash' method based on the amount of ash left behind from peat combustion [6,31] implies more assumptions, as part of the ash may be carried away from the burned area by wind and water erosion. In our case, the upper 5 cm (plot 6), 10 cm (plot 5), and 15 cm (plots 1 and 2) of the peat soil in all sections outside the fire area had a much higher ash content (up to 20% and more) compared with the underlying horizons (typically below 3–4%, Table 2), which must result from wind drifted ash from the burned area, i.e., ash from both biomass and peat combustion. Therefore, the reliability of the 'excess ash' method will rapidly decrease with time after the fire. For our method, the time that passed since the fire is of less importance.

Our estimates for the depth of soil burning after a forest-peat fire are similar to those published for peatlands in the forest zone [5,43–47]. They also confirm the spatial irregularity of peat soil burning, as noted by most authors. Our estimates of soil C losses are at the upper range of peat fire losses reported for North America (15–25 tC ha$^{-1}$) and Northern Eurasia (17–23 tC ha$^{-1}$) [48]. The (limited) number of studies on peat fire C losses

from boreal and temperate zones provided mean values of 22–28 tC ha$^{-1}$, although lower values have also been reported (11–15 tC ha$^{-1}$) [10]. We found such range (mean 4.3–28.7, median 22–29.7 tC ha$^{-1}$) in our earlier data [6].

The results of our study are thus not extraordinary. Other studies have also reported carbon losses from boreal and temperate zone peat fires reaching and even exceeding 100 tC ha$^{-1}$ [8,10,49]. The high soil C losses are, in our case, probably attributable to the vicinity of the road and the ditch remnants in the adjacent forested area, which would have contributed to the drainage of the site. A lower groundwater table allows the peat soil to burn deeper [5] with increased soil carbon loss [6]. Drainage also leads to shrinkage, compaction and an increase in carbon density of the peat. A comparison of undrained and drained sites showed that the depth of burning was 7 cm and 19 cm, respectively, but that the latter sites had lost nine times more carbon, reaching 170 tC ha$^{-1}$ [49]. Soil burning depth, carbon losses and carbon dioxide emissions to the atmosphere will be larger if the peatland water regime has been altered by human action or climate change, increasing their fire risk [6,7,50–52].

The estimated soil carbon losses from the entire burned area (mean 98 and median 92 tC ha$^{-1}$) significantly exceed the potential carbon losses from live biomass. Even after an intense forest fire on mineral soil, most tree trunks are preserved, at least at the root. By contrast, even low-intensity ground fires can have devastating consequences for a stand on peat soil by deepening the fire and turning it into a forest-peat fire, which damages the tree root systems [5].

The carbon loss of the tree stands biomass averaged 58.8 tC ha$^{-1}$ for the entire burned area, which is at least 1.5 times less than the estimated loss of soil carbon. Carbon loss from the combustion of ground vegetation and locally developed understorey was not considered, but its values will not change the overall picture significantly. Tree biomass carbon losses may vary significantly (up to four or more times) depending on the initial stand (Figure 7). Soil carbon losses, on the other hand, were fairly uniform despite the differences in site conditions.

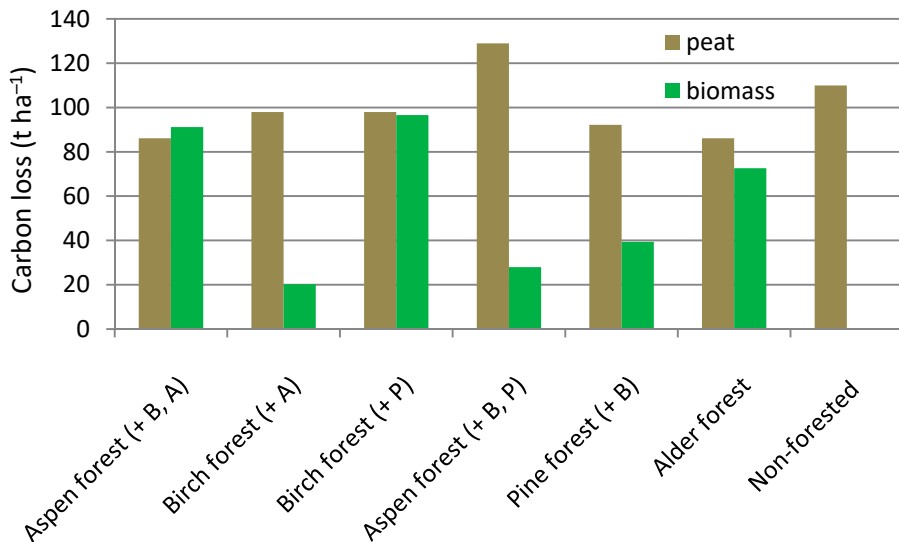

**Figure 7.** Biomass and peat carbon losses in sites with different woody vegetation. A, Alder; B, Birch; P, Pine.

In the case of this forest-peat fire, soil carbon loss may have caused a one-time release of almost 400 tCO$_2$ ha$^{-1}$. Even when disregarding the combustion of ground cover and part of the tree biomass, soil carbon loss alone contributes significantly to the carbon dioxide emissions into the atmosphere. For comparison, carbon losses from intensive milled peat extraction in this region were estimated at 25–32 tC ha$^{-1}$ year$^{-1}$ (=100–128 tCO$_2$ ha$^{-1}$ year$^{-1}$), CO$_2$ emission from bare peat on abandoned extraction fields at 6.4–18.8, and abandoned

hayfields on drained peatlands at 3.2 t $CO_2$ ha$^{-1}$ year$^{-1}$ [53]. When the loss of carbon from the tree biomass is considered, the total carbon dioxide emissions were almost 650 t$CO_2$ ha$^{-1}$ in our case. Carbon losses from fires are episodic but given their frequency (in the region under consideration, large-scale fires occurred; for example, in 1978, 2002, and 2010 [19]), their emissions may be the most important in the region.

## 5. Conclusions

Forest-peat fires are notable for their difficulty in estimating carbon losses. Our study of an 8 ha large forest-peat fire site in the Moscow region reconstructed the original characteristics of the destroyed forest stands by using pre-disturbance space imagery in combination with ground truthing surveys in adjoining areas. We also determined burn depth and soil carbon loss by reconstructing the pre-fire soil surface using the root collar of remaining stumps and comparing the peat characteristics from burned and adjacent unburned areas.

The mean (median) peat burn depth across the burned area was $15 \pm 8$ (14) cm, with differences across sites varying from $13 \pm 5$ (11) to $20 \pm 9$ (19) cm with a maximum of up to 50 cm. Burn depth increased with the relative height of the reconstructed pre-fire surface, probably due to better drainage, and was largest in areas dominated by aspen. These burn depth estimates are close to those from other peat fires in the forest zone and confirm the spatial irregularity of peat soil burning.

The layer-wise determination of peat bulk density, ash and carbon content allowed us to establish carbon losses as a function of soil burning depth. Soil carbon losses varied over the burned area from $9.22 \pm 3.75$ (mean) and 8.0 (median) to $11.0 \pm 4.96$ and 11.0 kg m$^{-2}$. Over the entire burned area, soil carbon loss was $9.8 \pm 5.57$ (mean) and 9.22 kg m$^{-2}$ (median), which is at the upper range of available values for peat fires in the boreal and temperate zones; values exceeding 100 tC ha$^{-1}$ have also been found in other studies. The high values in our case are attributable to the partial drainage of the site, which not only leads to deeper burning but also to soil compaction, which causes larger carbon losses per unit depth.

The estimated soil carbon loss over the entire burned area of 98 (mean) and 92 (median) tC ha$^{-1}$ significantly exceeds the potential carbon losses from biomass. The carbon loss of the tree stands biomass averaged 58.8 tC ha$^{-1}$ for the burned area, which is at least 1.5 times less than the estimated soil carbon loss. Carbon losses from the combustion of ground vegetation and locally developed understorey were disregarded but would not have affected the overall loss ratio significantly. The loss of tree biomass carbon differed substantially (up to 4 times or more) depending on the original stand. In contrast, soil carbon losses were fairly uniform, despite the variety in site conditions.

The loss of soil carbon in an underground (peat) forest fire equals a one-time release of nearly 400 t$CO_2$ ha$^{-1}$ into the atmosphere if we assume that most peat was burned to carbon dioxide, i.e., without considering other carbon-containing gases as well as black carbon. This value is 60% more than the potential $CO_2$ release from the burned tree stands biomass. Together the soil and biomass carbon losses caused carbon dioxide emissions of almost 650 t$CO_2$ ha$^{-1}$. The results confirm the underestimated impact of underground (peat) forest fires in the boreal zone on atmospheric $CO_2$ concentrations and climate compared with the tropics.

**Supplementary Materials:** The following are available online at http://www.mdpi.com/xxx/s1. Figure S1: General view of the investigated burned area, Table S1: Stand structure and composition of field plots, Table S2: Living biomass (dry matter) and carbon stock of tree stands on field plots, Figure S2: Results of determination of tree stump species and diameters, Figure S3: Site surface height before and after the fire, Figure S4: Distribution of the burn depths within the stand thematic classes in percentage of the class area, Table S3: Characteristics of peat on sampling plots, Figure S5: Peat burn depth (cm) vs. Relative height of the original soil surface before fire (cm) reconstructed from the root collar of stumps of trees of different species.

**Author Contributions:** A.S. designed and organized the research; A.M. conducted the forest cover study based on remote sensing and field data; D.M. conducted peat field and laboratory work; Y.G. provided forest biomass assessment; A.S., A.M., D.M. and H.J. analysed the results; and all authors contributed to writing the paper. All authors have read and agreed to the published version of the manuscript.

**Funding:** This study was supported by the project "Restoring peatlands in Russia—for fire prevention and climate change mitigation" financed under the International Climate Initiative (IKI) by the German Federal Ministry for the Environment, Nature Conservation, and Nuclear Safety (BMUB), facilitated through KfW, (project no. 11 III 040 RUS K. Restoring Peatlands), by the Russian Science Foundation (project 19-74-20185), and by the Russian Fund for Basic Research (project 15-05-07762).

**Data Availability Statement:** Not applicable.

**Acknowledgments:** The authors are grateful to Inga Gummert (Greifswald University) for active field and laboratory work, and Franziska Maier for field assistance. We thank Susanne Abel and Andreas Haberl from the Greifswald Mire Centre for their help in finding and selecting the object of research. We are grateful to Yu.A. Gopius and A.V. Markina (Institute of Forest Science of the Russian Academy of Sciences) for their help in field research; to O.N. Uspenskaya (Federal Scientific Vegetable Center/Institute of Forest Science of the Russian Academy of Sciences) for determining the botanical composition and degree of decomposition of peat; to G.G. Suvorov (Institute of Forest Science of the Russian Academy of Sciences) for determining the elemental composition of peat. The authors express their gratitude to the ScanEx R&D Centre for providing remote sensing data.

**Conflicts of Interest:** The authors declare no conflict of interest.

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
