# Peer review of "Assessing Wood and Soil Carbon Losses from a Forest-Peat Fire in the Boreo-Nemoral Zone"

_forests, doi:10.3390/f12070880_

Round 1
Reviewer 1 Report
This work presents a detailed survey about the depth of burn of a peat fire in Russia. In general, this is a useful research, and the results are valuable for the community. A few comments are provided.
- The value of the carbon emission flux (C/ha) in this work is based on the assumption that the entire region is burnt. This value could be mis-used by others, when scaling up the carbon loss emission to a large area, because there could be a large portion of the unburnt area. It would be useful to provide the data for the burning ratio, by defining some threshold value, which could be extracted in Fig. 5. Such data will be valuable to the community.
- The influence during the gap between the fire event in 2010 and the study in 2013 should be clarified more.
- The statement “Compared to the tropics, less data is available on emissions from peat fires in non-tropical regions of the world” is not appropriate. In fact, there are lots of emission data of fire in North America, Siberia, the British Isles, etc. On the opposite, tropical fire emission data are often only available in Southeast Asia, especially Indonesia, and these data are also limited.
- The estimation of carbon loss could be based on element analysis, but the emission of CO2 needs to consider the emission factor (EF). More data and discussion should be included.
- In the abstract, the method of using satellite data and field survey should be stated more clearly.
- Fig. 2 should include the scale bar
- Some abbreviations like DBH, should be explained in the beginning. Usually, the peat fire community and IPCC use the terminology of the depth of burn (DOB).
- The peat moisture content should be listed in the supplementary material, because it can be more important than the bulk density.
- The local weather and climate information (e.g., average temperature, precipitation) of the fire field should be listed.
Author Response
Dear Reviewer,
We appreciate your efforts in reviewing the manuscript and the important comments made. Responses to your comments are given below.
- The value of the carbon emission flux (C/ha) in this work is based on the assumption that the entire region is burnt. This value could be mis-used by others, when scaling up the carbon loss emission to a large area, because there could be a large portion of the unburnt area. It would be useful to provide the data for the burning ratio, by defining some threshold value, which could be extracted in Fig. 5. Such data will be valuable to the community.
We added a clarification in Section 2.1 that the model area in question was completely affected by the fire. When scaling up carbon loss emissions to large areas, the data obtained can be referred only to the burned areas, excluding those not affected by the fire.
Figure 5 shows differences in carbon loss within burned area, and in text we try to discuss these spatial differences. Table 2 presents the carbon losses with reference to the different tree cover classes we have identified. They show the level of possible peat fire carbon loss. These values, as well as the averages for the whole burned area, can be considered as some threshold values.
- The influence during the gap between the fire event in 2010 and the study in 2013 should be clarified more.
We have added a clarification to section 2.6. The results of peat root-neck burnout determination should not be greatly influenced by the time since the fire.
- The statement “Compared to the tropics, less data is available on emissions from peat fires in non-tropical regions of the world” is not appropriate. In fact, there are lots of emission data of fire in North America, Siberia, the British Isles, etc. On the opposite, tropical fire emission data are often only available in Southeast Asia, especially Indonesia, and these data are also limited.
We agree with the reviewer about the presence of emission data related to fires in North America, the British Isles, and Siberia. This is fully true for forest fires. But in this case we are talking exclusively about peat fires and we refer to the opinion of M. Turetsky and her coauthors who studied peat fires in different regions of the world (Nature Geoscience 2015, 8, 11–14, doi:10.1038/ngeo2325) and concluded that the ecology of peat fires and the role of peat fires in long-term Earth system processes need to be explored more thoroughly in future research.
- The estimation of carbon loss could be based on element analysis, but the emission of CO2 needs to consider the emission factor (EF). More data and discussion should be included.
We added a clarification at the end of the discussion. We did not set out to consider the emission coefficients resulting from a peat fire. The possible carbon dioxide emissions associated with the fire were given in the conclusion of the article were obtained by simply translating the carbon loss into carbon dioxide from the simple assumption that most of it would go exactly to its formation.
- In the abstract, the method of using satellite data and field survey should be stated more clearly.
This part has been rewritten, information was added.
- Fig. 2 should include the scale bar.
Done.
- Some abbreviations like DBH, should be explained in the beginning. Usually, the peat fire community and IPCC use the terminology of the depth of burn (DOB).
Done. DBH = (Tree’s) Diameter at Breast Height.
- The peat moisture content should be listed in the supplementary material, because it can be more important than the bulk density.
The peat moisture content of the peat, of course, differed from sample to sample. However, it was not required to calculate the carbon stock in the peat samples. The stock was calculated based on the bulk density, ash content, and C content.
- The local weather and climate information (e.g., average temperature, precipitation) of the fire field should be listed.
Climate information of the study region was added in section 2.1.
A technical typo has been corrected in the supplementary material.
Reviewer 2 Report
A very well-written paper, clearly establishing a global background, concisely establishing a useful picture of the landscape context, and describing the investigation.
Use of N in Section 2.7 and elsewhere for peat characterisation locations but without prefix for peat inventory plots in Figure 4 seems to refer to the same entities. If so, these should use a single nomenclature.
The linear relationship established between peat depth and carbon stock conceals the wider spread of values at greater depth, where the effect of variation on C stock estimates is greatest. If authors have any further justification for applying the linear relationship, or wish to identify this as a limitation, they might consider that.
Confirm colour patches in the legend of Figure 4 match those of the map.
The phrase "at least 1.5 times less than" (used twice) is ambiguous, and should be replaced by positive expression of amounts or proportions, such as "sixty percent of" or "about 40 tC ha-1 less than".
The citation of Davies et al. 2013 should be done in the journal style.
Comparing this episode of emission to continuing emissions on an annual basis for other land uses seems to understate their importance, given that they are ongoing every year. Certainly such comparisons are relevant to the reader, but increasing temporal or scale may be needed to make valid comparisons.
Supplementary material is excellent.
Overall, an excellent report of this investigation, effectively demonstrating the value of this widely-applicable low-tech method.
Author Response
Dear Reviewer,
We appreciate your efforts in reviewing the manuscript and the important comments made. Responses to your comments are given below.
Use of N in Section 2.7 and elsewhere for peat characterisation locations but without prefix for peat inventory plots in Figure 4 seems to refer to the same entities. If so, these should use a single nomenclature.
Corrected.
The linear relationship established between peat depth and carbon stock conceals the wider spread of values at greater depth, where the effect of variation on C stock estimates is greatest. If authors have any further justification for applying the linear relationship, or wish to identify this as a limitation, they might consider that.
Additional explanation added to the section
Confirm colour patches in the legend of Figure 4 match those of the map.
Yes, map and legend use the same colours.
The phrase "at least 1.5 times less than" (used twice) is ambiguous, and should be replaced by positive expression of amounts or proportions, such as "sixty percent of" or "about 40 tC ha-1 less than".
In accordance with your comment changes were made in Conclusions.
The citation of Davies et al. 2013 should be done in the journal style.
Corrected.
Comparing this episode of emission to continuing emissions on an annual basis for other land uses seems to understate their importance, given that they are ongoing every year. Certainly such comparisons are relevant to the reader, but increasing temporal or scale may be needed to make valid comparisons.
In accordance with the reviewer's comment an addition is made at the end of Discussion.
A technical typo has been corrected in the supplementary material.